# Homocysteine and Inflammatory Cytokines in the Clinical Assessment of Infection in Venous Leg Ulcers

**DOI:** 10.3390/antibiotics11091268

**Published:** 2022-09-18

**Authors:** Ilaria Cavallo, Ilaria Lesnoni La Parola, Francesca Sivori, Luigi Toma, Tatiana Koudriavtseva, Isabella Sperduti, Daniela Kovacs, Giovanna D’Agosto, Elisabetta Trento, Norma Cameli, Anna Mussi, Alessandra Latini, Aldo Morrone, Fulvia Pimpinelli, Enea Gino Di Domenico

**Affiliations:** 1Microbiology and Virology, San Gallicano Dermatological Institute, IRCCS, 00144 Rome, Italy; 2Department of Dermatology, San Gallicano Dermatological Institute, IRCCS, 00144 Rome, Italy; 3Department of Research, Advanced Diagnostics, and Technological Innovation, Translational Research Area, IRCCS Regina Elena National Cancer Institute, 00144 Rome, Italy; 4Medical Direction, IRCCS Regina Elena National Cancer Institute, 00144 Rome, Italy; 5Biostatistics, IRCCS Regina Elena National Cancer Institute, 00144 Rome, Italy; 6Cutaneous Physiopathology, San Gallicano Dermatological Institute, IRCCS, 00144 Rome, Italy; 7Scientific Direction, San Gallicano Institute, IRCCS, 00144 Rome, Italy; 8Department of Biology and Biotechnology “C. Darwin”, Sapienza University, 00185 Rome, Italy

**Keywords:** leg ulcers, venous ulcer, biofilm, homocysteine, infection, wound, cytokines

## Abstract

Inflammation and biofilm-associated infection are common in chronic venous leg ulcers (VU), causing deep pain and delayed healing. Albeit important, clinical markers and laboratory parameters for identifying and monitoring persistent VU infections are limited. This study analyzed 101 patients with infected (IVU) and noninfected VUs (NVU). Clinical data were collected in both groups. The serum homocysteine (Hcys) and inflammatory cytokines from the wound fluid were measured. In addition, microbial identification, antibiotic susceptibility, and biofilm production were examined. IVU were 56 (55.4%) while NVU were 45 (44.5%). IVUs showed a significant increase in the wound’s size and depth compared to NVUs. In addition, significantly higher levels of interleukin (IL)-6, IL-10, IL17A, and tumor necrosis factor-alpha (TNF-α) were found in patients with IVUs compared to those with NVUs. Notably, hyperhomocysteinemia (HHcy) was significantly more common in patients with IVUs than NVUs. A total of 89 different pathogens were identified from 56 IVUs. Gram-negative bacteria were 51.7%, while the Gram-positives were 48.3%. At the species level, *Staphylococcus aureus* was the most common isolate (43.8%), followed by *Pseudomonas aeruginosa* (18.0%). Multidrug-resistant organisms (MDROs) accounted for 25.8% of the total isolates. Strong biofilm producers (SBPs) (70.8%) were significantly more abundant than weak biofilm producers (WBP) (29.2%) in IVUs. SBPs were present in 97.7% of the IVUs as single or multispecies infections. Specifically, SBPs were 94.9% for *S. aureus*, 87.5% for *P. aeruginosa*, and 28.6% for *Escherichia coli*. In IVU, the tissue microenvironment and biofilm production can support chronic microbial persistence and a most severe clinical outcome even in the presence of an intense immune response, as shown by the high levels of inflammatory molecules. The measurement of local cytokines in combination with systemic homocysteine may offer a novel set of biomarkers for the clinical assessment of IVUs caused by biofilm-producing bacteria.

## 1. Introduction

The presence of a chronic venous leg ulcer (VU) is a debilitating condition highly prevalent in the elderly population. In European countries and the United States, approximately 2% of the entire population is affected by chronic wounds, with VUs accounting for up to 70% of lower leg ulcers [1,2,3]. VU is an open wound between the knee and the ankle joint that remains unhealed for more than 30 days due to venous insufficiency [4]. The prolonged inflammation observed in the VUs leads to leukocytes’ recruitment and the release of inflammatory modulators that negatively affect the surrounding tissues [5]. Although the wound healing process involves inflammation and resolution of the inflammatory response, the excessive production of inflammatory mediators may cause deleterious consequences [6]. Previous studies have identified up-regulation of cytokines in patients with VU, suggesting that chronic venous hypertension results in an inflammatory state responsible for poor wound healing [7,8]. In addition, inflammatory markers such as interleukin (IL)-1, IL-6, IL-10, and tumor necrosis factor-alpha (TNF-α) have been shown to significantly increase in VU than in acute healing wounds [9,10,11,12]. Similar results have been observed for angiogenic factors such as the Vascular Endothelial Growth Factor (VEGF), epidermal growth factor receptor (EGFR), basic fibroblast growth factor (bFGF), and other cytokines [13,14]. However, no consensus exists for sampling procedures or measuring systems, and the large heterogeneity of cytokine levels has limited the potential benefit of previous research [15].

Elevated serum homocysteine (Hcy), defined as hyperhomocysteinemia (HHcy), is an independent risk factor for thrombovascular diseases [16,17]. It has been observed that in the presence of HHcy, the Hcy levels within a lesion are significantly higher than the serum levels. This condition leads to T cells and monocytes activation with T cells differentiation into a proinflammatory Th1 phenotype that, in turn, activates the local macrophage, smooth muscle, and foam cell populations [18]. The levels of Hcy within a tissue microenvironment may also influence the local immune response leading to either enhanced inflammation by immune cell activation and alterations in the cytokine profiles or diminished responsiveness through cellular apoptosis and death [19,20,21,22]. Although evidence demonstrated clear associations between elevated Hcy and vascular risks, the specific mechanisms by which HHcy contributes to vascular diseases remain uncertain. Nevertheless, in VUs, the impaired host immune response, necrotic tissues, and debris promote bacterial attachment and infection. A VU fails to heal in the event of infection and can be recalcitrant to conventional antimicrobial treatment due to microbial biofilm [23]. Indeed, biofilm acts as an important predisposing factor to the chronicity of a nonhealing VU by providing a protective environment against the host immune response and antimicrobial penetration [24,25]. Thus, aggressive and prolonged antibiotic treatment, based on conventional susceptibility testing, may show poor efficacy in eradicating biofilm infections since the minimum bactericidal concentration cannot be achieved in vivo without posing severe adverse effects to the host [26]. Furthermore, the biofilm lifestyle increases the horizontal transfer of virulence genes and the development of multidrug-resistant organisms (MDRO) [27,28]. Therefore, biofilms significantly impact VU’s treatment and must be considered when developing a targeted treatment plan. This study analyzes putative factors for discriminating between IVU versus NVU and the contribution of biofilm-growing bacteria in patients with VU to promote the early detection of high-risk subjects and prevent further complications.

## 2. Results

From January 2019 to April 2021, 101 patients with infected (IVU; *N* = 56) and noninfected chronic venous leg ulcers (NVU; *N* = 45) were enrolled in the study. The demographic and clinical characteristics are summarized in Table 1. IVUs showed a significant increase in size (*p* < 0.0001), and depth (*p* = 0.0008) of the ulcer as compared to NVUs. Notably, a significantly (*p* < 0.0001) higher number of patients with hyperhomocysteinemia (HHcy) was reported in IVU than in NVU. In addition, significantly higher levels of IL-6 (*p* = 0.0003), IL-10 (*p* = 0.02), IL17A (*p* < 0.0001), and TNF-α (*p* = 0.0003) were found in patients with IVUs compared to those with NVUs. Conversely, interferon-gamma (IFN-γ) was the only inflammatory molecule significantly (*p* = 0.03) higher in NVU than IVU.

Before systemic antibiotic treatment, samples were collected from IVUs for microbiologic analysis. A total of 89 different pathogens were identified from 56 IVUs. In particular, single bacterial species were detected from 32 ulcerative lesions, while two different species were observed in 16 lesions. In addition, three different microorganisms were observed in seven cases, and four bacterial species were isolated in one patient.

Gram-negative bacteria accounted for 51.7% (*n* = 46), while the Gram-positives were 48.3% (*n* = 43) of the total. At the species level, *Staphylococcus aureus* was the most common isolate (43.8%; *n* = 39), followed by *Pseudomonas aeruginosa* (18.0%; *n* = 16). In descending order, other relevant pathogens were *Escherichia coli* (7.9%; *n* = 7), *Serratia marcescens* (6.7%; *n* = 6) and *Enterobacter cloacae* (3.4%; *n* = 3) and *Morganella morganii* (Table 2).

The antimicrobial susceptibility profiles of Gram-negative and Gram-positive isolates are summarized in Figure 1A,B. *S. aureus* showed high resistance to fluoroquinolones (ciprofloxacin and levofloxacin) and aminoglycosides (gentamicin and tobramycin). No decrease in susceptibility to dalbavancin, daptomycin, linezolid, tigecycline, and vancomycin was detected in the *S. aureus* isolates. *P. aeruginosa* showed resistance to ampicillin/sulbactam, cefepime, ceftazidime, ciprofloxacin, levofloxacin and piperacillin/tazobactam and was totally susceptible only to colistin and ceftolozane/tazobactam. Carbapenems, colistin, ampicillin/sulbactam, ceftazidime/avibactam, ceftolozane/tazobactam and piperacillin/tazobactam were the most active antibiotics against *E. coli*.

Of the 89 isolates, 25.8% (*n* = 23) were classified as MDROs including methicillin-resistant *S. aureus* (MRSA) (*n* = 13), multidrug-resistant *P. aeruginosa* (*n* = 4), extended-spectrum β-lactamase (ESBL)-producing *E. coli* (*n* = 5) and *S. maltophilia* resistance to trimethoprim/sulfamethoxazole.

Bacterial biofilm plays a major role in delaying chronic wound healing resulting in a significant health care burden. Thus, we measured the level of biofilm production in all the isolates (Figure 2). Strong biofilm producers (SBPs) (70.8%) were more abundant than weak biofilm producers (WBP) (29.2%) in IVUs. At the single species level, SBPs were detected in 94.9% of *S. aureus*, 87.5% of *P. aeruginosa*, 28.6% of *Escherichia coli*, and 16.7 of *S. marcescens* isolates. Notably, SBPs were present in 96.4% (*N* = 54) of the IVUs as single or multispecies infections. Conversely, WBPs were found only in two cases of IVUs (13.3%) as single bacterial species.

Fluorescence microscopy analysis of *S. aureus*, *P. aeruginosa*, and *E. coli*, classified as SBP (Figure 2B), showed the presence of a uniform layer of mature biofilm ranging from 20 to 60 μm with different morphologies after 24 h of incubation.

## 3. Discussion

VUs are prone to bacterial infection, leading to serious complications. Managing VUs in the presence of an infection is still challenging due to prolonged healing and frequent reoccurrence [29,30]. The diagnosis of infection in a VU remains problematic and debated among health professionals [29]. This study showed that Gram-negative bacteria were 51.7%, mostly represented by *P. aeruginosa*, followed by *E. coli*, *S. marcescens*, *E. cloacae*, and *M. morganii*. Gram-positives were 48.3%, with *S. aureus* emerging as the dominant species. These data are in agreement with other reports from patients with VUs showing comparable microbial distribution [31,32,33]. Historically, culture-based methods represented the gold standard for microbial isolation and ulcer identification [34]. However, the culture-independent molecular method has revealed greater bacterial diversity and load than traditional culture-based methods [35,36,37,38]. Likewise, another study of chronic wounds of various etiologies revealed that aerobic cultures significantly underestimate the number of bacterial taxa compared with 16S rRNA sequencing [37]. Specifically, nine of the 20 most represented bacteria identified with 16S rRNA sequencing were anaerobes that escaped with aerobic culture methods [37]. Indeed, anaerobes have been implicated in wound chronicity and biofilm production [39,40], and the presence of anaerobic bacteria has been reported in pressure ulcers [41,42]. Thus, analyzing skin microbiota composition in VUs is central to identifying novel therapeutic strategies for patients with chronic wounds [35,43]. However, despite the 16S rRNA sequencing addresses many of the limitations of culture-based approaches, the active contribution of different bacterial species to wound outcomes can be addressed only by culture-based approaches [35,36,38]. Bacterial biofilms growing on the surface of chronic wounds, such as VUs, are implicated in promoting chronic inflammation, which leads to delayed healing [40,44,45,46]. In recent years, the wound infection continuum has included microbial biofilm and its role in wound infection [47,48,49]. This study demonstrates that SBPs were significantly more abundant than WBP on the surface of chronic wounds. Multiple bacterial species from wound specimens is common in many studies [50]. Other reports showed that a chronic wound biofilm comprises multiple bacterial species held together within the biofilm matrix [51]. Specifically, we identified 89 bacterial strains from 56 IVUs. In particular, single bacterial species were detected from 32 ulcerative lesions, while two different species were observed in 16 lesions. In addition, three different microorganisms were observed in seven cases, and four bacterial species were isolated in one patient. Notably, bacterial strains classified as WBP were in most cases associated with polymicrobial infections in association with SBP, confirming previous observations [52]. This observation suggests biofilm represents a key virulence determinant in promoting bacterial persistence and chronicity of ulcerative lesions [52,53]. Excessive inflammation plays a major role in delayed tissue regeneration and wound healing. However, until now, routine laboratory parameters are not available to assess the inflammatory status of VUs. Elevated inflammatory cytokine levels compared to healthy tissue have been implicated in the pathogenesis of non-healing VU [11,54]. Consistent with other studies, our data also indicates the presence of a robust pro-inflammatory environment in VU. Moreover, the present data showed a significant increase in IL-6, IL-10, IL17A, and TNF-α concentrations in wound fluids of patients with IVU compared with NVU. A pilot study showed a significant increase of IL-6 and TNF-α from wound fluids of patients with mixed bacterial infections compared with monomicrobial-infected wounds [55]. A previous study reported that the pro-inflammatory cytokines IL-1, IL-6, and TNF-α have significantly higher levels in the fluid of non-healing wounds compared with healing chronic leg ulcers [11]. Staphylococcal peptidoglycan or lipopolysaccharide (LPS) are potent triggers of cytokine release, and IVUs are constantly exposed to various bacterial stimuli. Microbial endo and exoproducts released by pathogenic bacteria can trigger various cytokine production pathways to promote their survival and persistence within the host [56]. In particular, deregulation of the potent anti-inflammatory cytokine IL-10 is increasingly associated with such virulence strategies leading to poor antimicrobial effector mechanisms, increased disease severity, and chronicity in *S. aureus* and *P. aeruginosa* infections [57,58,59]. Since cytokines can also be influenced by ulcer size, prospective studies are needed further to validate their clinical application as a diagnostic tool. Although the pathogenesis of VU is likely multi-factorial, our results indicate that the increase of specific inflammatory mediators, such as IL-6, IL-10, IL17A, and TNF-α in wound fluids can be found in the presence of an infection particularly when sustained by biofilm-producing bacteria.

In contrast to local cytokine levels, the serological marker CRP was not significantly associated with the presence of any of the investigated conditions. This reinforces previous observations suggesting that CRP cannot be used as a diagnostic marker of tissue infections without bone involvement or to discriminate between colonization, critical colonization, and local infection [55,60].

In the present study, we also documented that HHcy was significantly more common in patients with IVU compared with NVU. HHcy is considered an independent risk factor for cardiovascular and neurocognitive diseases [20,21,61,62] contributing to developing endothelial dysfunction [63]. Others reported the direct association of diabetic foot ulcers with increased plasma Hcy levels [64]. More recently it has been reported that elevated serum Hcy levels inhibit endothelial cell wound repair and angiogenesis [65,66,67,68,69]. Experimental evidence also suggests that increased Hcy levels promote vascular oxidative stress, activating a series of chemical reactions that lead to inflammation of the arterial wall and progression of the atherosclerotic plaque [70]. In eukaryotic cells, Hcy has three main metabolic fates: methionine production, cysteine biosynthetic pathway, or being released into the extracellular medium. At the same time, Hcy is a crucial intermediate for *de novo* synthesis of methionine and in a quorum-sensing (QS) mechanism based on the autoinducer-2 (AI-2) both in Gram-positive and Gram-negative bacteria. Notably, in bacterial pathogens such as *P. aeruginosa* and *Enterococcus faecalis*, AI-2 regulates the production of virulence factors and biofilm formation [71,72,73,74]. From our data, one could speculate on a possible contribution of serum Hcy as a potential signaling molecule for the resident biofilm community in the ulcer. However, because its pathophysiology is still unclear, more studies are needed to clarify the role of Hcy in developing VU and the relationship between an infected ulcer and Hcy levels.

Several limitations must be noted when considering the data produced in this study. First, our microbiological analysis was restricted to aerobic bacteria. Although anaerobes have been identified as a significant contributor to wound bioburden, standardized culturing and identifying procedures in clinical microbiology laboratories are uncommon for VU [75]. However, widespread opinion among wound care practitioners is that aerobic or facultative pathogens are the primary causes of delayed healing and infection in acute and chronic wounds [76]. Moreover, the relevance of wound swab sampling has been repeatedly questioned in favor of biopsied tissue, suggesting that the associated microflora does not reflect that of deeper tissue [77]. However, it should be noted that wound contamination/infection commonly occurs from sources external to the wound. Thus, it is highly unlikely that superficial tissue will result sterile while deeper tissue is infected [76].

In summary, these data further confirmed that studying bacterial biofilm within the wound is critical to understanding its pathophysiology and developing innovative diagnostic tools and targeted therapies. In addition, serological markers such as Hcy associated with the local cytokines’ detection may have a significant role in IVU identification. In particular, the assessment of these biomarkers could be effective in monitoring patients’ follow-up during antibiotic treatments, to evaluate the effectiveness of an ulcer debridement or other approaches aimed at reducing the bacterial load in patients with VUs. Thus, introducing new predictors potentially associated with VU may facilitate the early detection of high-risk subjects. Nevertheless, more prospective studies and higher resolution diagnostic tools may provide a better evidence-based diagnosis of VU.

## 4. Material and Methods

### 4.1. Patients and Samples

All patients with VUs admitted to the San Gallicano Dermatological Institute, Istituti Fisioterapici Ospitalieri (IFO) from March 2019 to November 2021 were included in the study. A VU was classified as chronic when it persisted for at least three months [78]. VUs were examined by a team of expert wound care specialists and classified as being infected or noninfected based on clinical signs (e.g., cellulitis, edema, erythema, heat, increased exudate, and pain) and a comprehensive review of medical records and laboratory test results [29,79]. IVUs that required systemic antibiotic treatments were sampled for microbiologic analysis before antibiotic administration. Sample collection was performed by commercially available swabs (COPAN swabs, Brescia, Italy) by rotating a sterile, premoistened swab across the surface of a leg ulcer according to departmental guidelines. Only one swab per patient was collected after careful wound cleaning and debridement to prevent surface contamination. All specimens were appropriately labeled, packaged, and processed within two hours after collection for culture analysis and biofilm assessment [52]. Exclusion criteria were the presence of fungi in a polymicrobial infection and a systemic antimicrobial treatment or local antiseptic therapy within two weeks before sampling. 

### 4.2. Microbiology

Bacterial identification was performed by matrix-assisted laser desorption/ionization-time of flight mass spectrometry (MALDI-TOF MS) system (Bruker Daltonik, Bremen, Germany). The antimicrobial susceptibility was assessed by the BD Phoenix^TM^ automated microbiology system (Becton Dickinson Diagnostic Systems, Sparks, MD, USA). Susceptibility for ceftazidime/avibactam, ceftolozane/tazobactam, colistin, and dalbavancin was determined by the Sensititre broth microdilution method (Thermo Scientific, Branchburg, NJ, USA), and results were interpreted according to the European Committee on Antimicrobial Susceptibility Testing (EUCAST) clinical breakpoints (http://www.eucast.org/clinical_breakpoints (accessed on 5 August 2022)) [80]. MDROs were defined as described previously [52] and according to the Healthcare Infection Control Practices Advisory Committee (HICPAC, https://www.cdc.gov/hicpac/index.html (accessed on 5 August 2022)) guidelines and the local infection control committee.

### 4.3. Biofilm Production

Bacterial isolates were analyzed for their ability to produce biofilm by the clinical BioFilm Ring Test (cBRT) (Biofilm Control, Saint Beauzire, France) as described previously [81,82]. Briefly, an overnight culture grown on a Chocolate agar plate was used to inoculate 2 mL of 0.45% saline solution to 1.0 ± 0.3 McFarland turbidity standard. The bacterial suspension was used to inoculate a 96-well polystyrene plate with 200 μL/well. The test was performed using the toner solution (TON004) containing magnetic beads 1% (*v*/*v*) mixed in the Brain Heart Infusion medium. Ten-fold serial dilutions were performed in a volume of 200 μL BHI/TON mix. *Staphylococcus epidermidis* ATCC 12228 were included in each plate as standard reference and internal control. After 5 h of incubation at 37 °C in a static condition, wells were covered with contrast liquid, placed for 1 min on the block carrying 96 mini-magnets, and scanned with a plate reader (Pack BIOFILM, Biofilm Control, Saint Beauzire, France). The adhesion strength of each strain was expressed as BioFilm Index [83] and used to classify strains as weak and strong biofilm producers [84]. Each strain was analyzed in duplicate, and experiments were repeated three times.

### 4.4. High-Sensitivity C-Reactive Protein (CRP) Assay

According to routine clinical practice, serum CRP level was measured on a Cobas c503 analyzer (Roche Diagnostics, Mannheim, Germany).

### 4.5. Wound Fluid Sampling and Cytokines Analysis

Wound fluid of 70 μL was collected from the ulceration site with commercially available nylon-flocked swabs (Minitip Flocked Swabs 551C; COPAN swabs, Brescia, Italy) after sharp debridement [85]. These swabs consist of perpendicular nylon fibers allowing liquid samples to uptake by capillary action. Subsequently, wound fluid was recovered from the swab by immediate centrifugation (10,000 rpm for 3 min at room temperature). Finally, all samples were cleared by centrifugation at 10,000× *g* for 20 min at four °C before cytokines analysis to obtain cell-free supernatants.

A broad range of immune mediators, including Interferon-gamma (IFN-γ), Interleukin (IL)-1β, IL-6, the chemokine IL-8, the regulatory cytokine IL-10, the T-cell–associated cytokines IL-17A, IL22, and the tumor necrosis factor-alpha (TNF-α) were measured in supernatants samples by using an automated ELISA assay Simple Plex™ technology on ELLA platform (Bio-Techne, Minneapolis, MN, USA) following the manufacturer’s instructions.

### 4.6. Biofilm Imaging

Bacterial isolates were grown overnight on blood agar plates. Single colonies were used to inoculate 3 mL of 0.45% saline solution (Air Life, Carefusion, CA, USA) to obtain turbidity of 0.5 ± 0.3 McFarland turbidity standard. Samples were diluted at 1:1000 and resuspended in 1 mL of BHI in a μ-Slide, eight well chamber slides (Ibidi, Gräfelfing, Germany). The bacterial suspension was incubated at 37 °C for 24 h to allow biofilm formation. Subsequently, the medium was removed, and samples were washed in a 0.45% saline solution. The biofilm cells were stained by the LIVE/DEAD BacLight kit (Life Technologies, New York, NY, USA) and examined with an Apotome system (Zeiss, Oberkochen, Germany) connected to an Axio Observer inverted fluorescence microscope (Zeiss). Data were analyzed with the ZEN 3.2 (blue edition) software (Zeiss) [80].

### 4.7. Statistics

All variables were summarized with descriptive statistics. The comparisons between continuous variables were made with the Student’s *t*-test or the Mann-Whitney U test, when appropriate. In contrast, categorical variables were tested using the χ^2^ or two-tailed Fisher’s exact test when appropriate. A *p* value of <0.05 was considered statistically significant. Statistical analyses were performed using SPSS software version 21 (SPSS, Inc., Chicago, IL, USA).

## Figures and Tables

**Figure 1 antibiotics-11-01268-f001:**
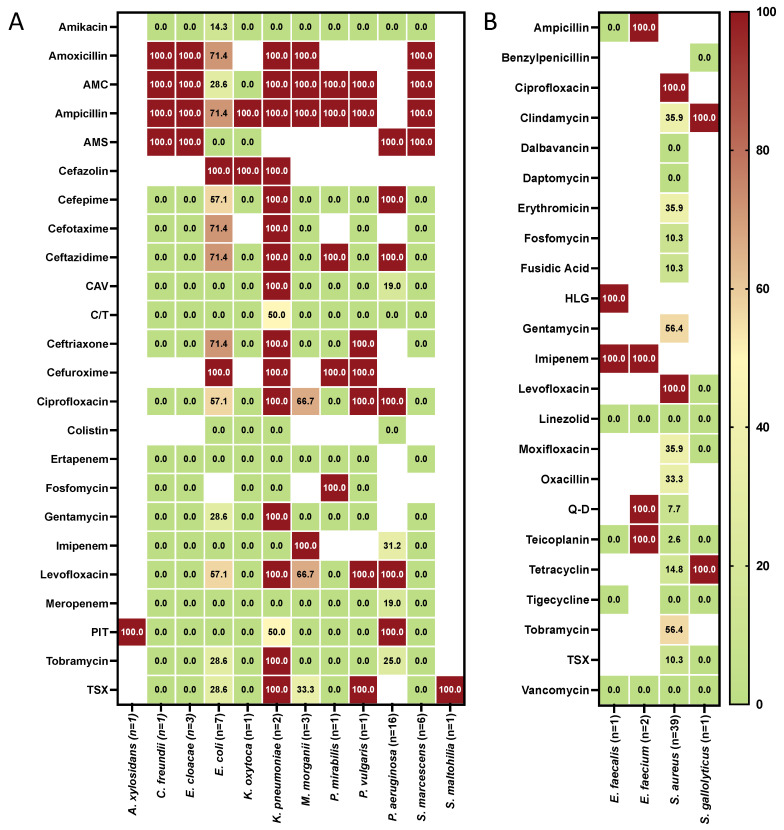
Antibiotic resistance pattern (% of resistance) for the indicated antibiotics tested against the Gram-negative (**A**) and Gram-positive (**B**) pathogens isolated from 56 patients with infected chronic venous leg ulcers. The numbers in brackets indicate the number of bacterial isolates. Isolates were classified as susceptible or resistant according to the European Committee on Antimicrobial Susceptibility Testing clinical breakpoint tables (EUCAST clinical breakpoint). Blank: not tested. (AMC) amoxicillin/clavulanic acid; (AMS) ampicillin-sulbactam; (CAV) ceftazidime-avibactam; (C/T) ceftolozane/tazobactam; (HLG) high-level gentamicin; (PIT) piperacillin/tazobactam; (Q-D) quinupristin-dalfopristin (TSX) trimethoprim/sulfamethoxazole.

**Figure 2 antibiotics-11-01268-f002:**
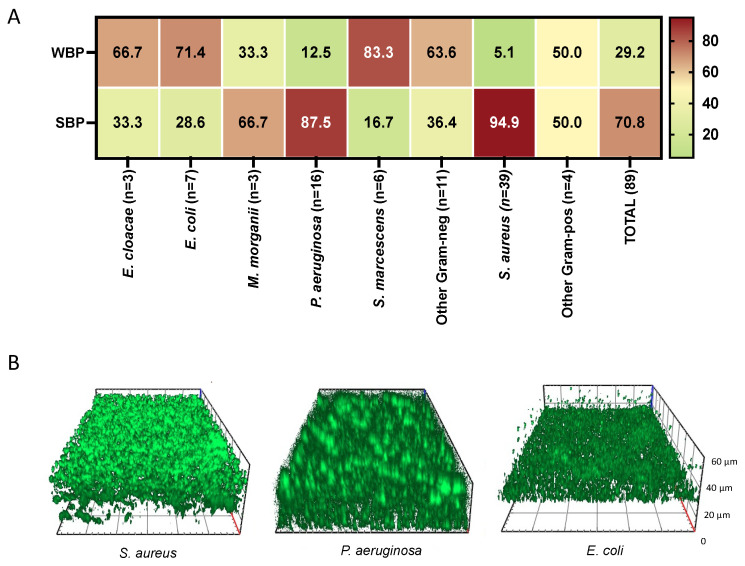
Biofilm production of bacterial isolates from patients with infected chronic venous leg ulcers. (**A**) Biofilm formation was assessed by the cBRT. Clinical isolates were classified as strong biofilm producers (SBPs) and weak biofilm producers (WBP). All results are expressed as a percentage. (**B**) Representative images of *Staphylococcus aureus*, *Pseudomonas aeruginosa*, and *Escherichia coli* isolates grown for 24 h at 37 °C. Orthogonal sections displaying horizontal and side views of reconstructed three-dimensional (3D) biofilm images are shown.

**Table 1 antibiotics-11-01268-t001:** Body mass index (BMI), hyperhomocysteinemia (HHcy) cut-off was set at >15.0 μmol/L; high-sensitivity C-reactive protein (CRP) cut-off was set at <1.0 mg/L. Interferon-gamma (IFN-γ), interleukin (IL)-1β, IL-6, IL-8, IL-10, IL17A, IL22 and tumor necrosis factor alpha (TNF-α).

Clinical Data	NVU (*N* 45)	IVU (*N* 56)	*p*
**Sex (Male/Female) (%)**	38/62	42/58	0.79
**Age (Years) (Median-Range)**	81 (41–97)	76.5 (28–97)	0.35
**BMI (Kg/m^2^) (Median-Range)**	26.7 (21.0–40.2)	25.2 (20.2–42.5)	0.78
**Diabetic (%)**	*N* = 6 (13.3)	*N* = 11 (20.4)	0.46
**Duration of ulcer (months) (Median-Range)**	6.8 (3–48)	4 (3–48)	0.72
**Size of ulcer (cm^2^) (Median-Range)**	8.6 (1.5–53)	28 (3–140)	**<0.0001**
**Depth of ulcer (% Grade 3)**	*N* = 2 (4.4)	*N* = 17 (30.3)	**0.0008**
**HHcy (% of positive subjects)**	*N* = 7 (15.6)	*N* = 33 (58.9)	**<0.0001**
**CRP (% of positive subjects)**	*N* = 28 (62.2)	*N* = 38 (67.9)	0.58
**IFN-** γ **(pg/mL) (Median-Range)**	1.44 × 10^1^ (1.11 × 10^0^–3.76 × 10^2^)	1.78 × 10^0^ (1.02 × 10^0^–8.21 × 10^1^)	**0.03**
**IL-1** β **(pg/mL) (Median-Range)**	7.49 × 10^4^ (3.73 × 10^3^–6.99 × 10^5^)	7.53 × 10^4^ (3.95 × 10^4^–4.93 × 10^5^)	0.80
**IL-6 (pg/mL) (Median-Range)**	1.84 × 10^4^ (4.71 × 10^2^–8.28 × 10^4^)	1.75 × 10^5^ (6.94 × 10^4^–3.31 × 10^5^)	**0.0003**
**IL-8 (pg/mL) (Median-Range)**	8.21 × 10^4^ (2.07 × 10^4^–8.94 × 10^5^)	1.30 × 10^5^ (3.89 × 10^4^–1.86 × 10^6^)	0.23
**IL-10 (pg/mL) (Median-Range)**	7.36 × 10^1^ (4.61 × 10^0^–3.12 × 10^2^)	1.42 × 10^2^ (6.26 × 10^1^–6.02 × 10^2^)	**0.02**
**IL-17A (pg/mL) (Median-Range)**	6.46 × 10^1^ (2.38 × 10^0^–2.01 × 10^2^)	9.50 × 10^3^ (5.49 × 10^1^–7.84 × 10^3^)	**<0.0001**
**IL-22 (pg/mL) (Median-Range)**	2.72 × 10^2^ (1.05 × 10^0^–8.34 × 10^2^)	2.47 × 10^2^ (1.03 × 10^0^–1.67 × 10^4^)	0.14
**TNF-** α **(pg/mL) (Median-Range)**	3.75 × 10^2^ (8.68 × 10^1^–1.81 × 10^3^)	1.59 × 10^3^ (3.94 × 10^2^–9.38 × 10^3^)	**0.0002**

**Table 2 antibiotics-11-01268-t002:** Bacterial isolates from patients with colonized skin ulcers.

Strain	*N*	%
*Staphylococcus aureus*	39	43.8
*Pseudomonas aeruginosa*	16	18.0
*Escherichia coli*	7	7.9
*Serratia marcescens*	6	6.7
*Enterobacter cloacae*	3	3.4
*Morganella morganii*	3	3.4
*Enterococcus faecium*	2	2.3
*Klebsiella pneumoniae*	2	2.3
*Shewanella algae*	2	2.3
*Achromobacter xylosoxidans*	1	1.1
*Bordetella trematum*	1	1.1
*Citrobacter freundii*	1	1.1
*Enterococcus faecalis*	1	1.1
*Klebsiella oxytoca*	1	1.1
*Proteus mirabilis*	1	1.1
*Proteus vulgaris*	1	1.1
*Stenotrophomnas maltophilia*	1	1.1
*Streptococcus gallolyticus*	1	1.1
**Total**	**89**	**100**

## Data Availability

The study data are available on reasonable request.

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
