# Peer review of "Homocysteine and Inflammatory Cytokines in the Clinical Assessment of Infection in Venous Leg Ulcers"

_antibiotics, 2022, doi:10.3390/antibiotics11091268_

Round 1

Reviewer 1 Report

The title is inappropriate as inflammatory parameters will always rise in an infected tissue irrespective of whether the bacteria is present in planktonic or biofilm form. Measuring the biofilm-forming ability of the bacterial isolates does not confirm that they are in biofilm form in wound tissue.

The introduction is too lengthy and needs to be curtailed.

In the result section, the following information needs to be added:

Table 2 can be omitted, and details can be added to the text.

What criteria were followed for classifying the bacteria as MDR?

On what basis were bacteria classified as strong/weak biofilm producers?

The discussion needs to be shortened as, at times, it appears as if one is reading a review on the subject. The statement that 89 different pathogenic bacteria from 56IVUs were isolated is incorrect (Line 205) as they are different strains of multiple bacteria but not 89 different bacteria (as the sentence suggests)

In the material and methods section, the clinical Biofilm Ring Test (cBRT) should be described.

No clear-cut recommendation for clinical assessment of VU patients has been made. 

Author Response

First, we would like to thank the reviewers for the time spent reviewing our manuscript and the valuable comments and suggestions, which were of great help in improving our work. Accordingly, the manuscript has been revised with new information and additional interpretation of the data. In particular, we have expanded the description of the Biofilm Ring Test and the definition of strong and weak biofilm producers. Besides, we have addressed all specific comments, as seen in the enclosed point-by-point list. The changes from the original submission are highlighted in a track changes version of the document.

Comments and Suggestions for Authors

The title is inappropriate as inflammatory parameters will always rise in an infected tissue irrespective of whether the bacteria is present in planktonic or biofilm form. Measuring the biofilm-forming ability of the bacterial isolates does not confirm that they are in biofilm form in wound tissue.

Reply: We agree with the reviewer that the in vitro condition used to measure biofilm production may not reflect the real in vivo conditions observed in a chronic leg ulcer. For this reason, we only referred to the potential ability to form a biofilm of different clinical isolates. However, in the revised manuscript, the title has been changed as suggested by the reviewer.

The introduction is too lengthy and needs to be curtailed.

R: The introduction has been shortened as suggested.

In the result section, the following information needs to be added:

 Table 2 can be omitted, and details can be added to the text.

R: We would prefer to maintain Table 2 and avoid introducing a long list of bacterial strains and data in the text to help the readers.

 What criteria were followed for classifying the bacteria as MDR?

R: The criteria were specified in the text (lines 348 - 351).

 On what basis were bacteria classified as strong/weak biofilm producers?

R: This section has been expanded to clarify the classification of strong/weak biofilm producers (lines 355 – 375) in the revised version of the manuscript.

The discussion needs to be shortened as, at times, it appears as if one is reading a review on the subject. The statement that 89 different pathogenic bacteria from 56 IVUs were isolated is incorrect (Line 205) as they are different strains of multiple bacteria but not 89 different bacteria (as the sentence suggests)

R: The discussion section has been shortened. Besides, the sentence in line 205 has been modified as highlighted by the reviewer.

In the material and methods section, the clinical Biofilm Ring Test (cBRT) should be described.

R: As stated before, this section has been expanded to include the description of the clinical Biofilm Ring Test (cBRT) procedure in detail and the classification of strong/weak biofilm producers (lines 355 – 375).

No clear-cut recommendation for clinical assessment of VU patients has been made. 

R: We thank the reviewer for pointing this out. However, we did not provide clear-cut recommendations for clinical assessment of VU patients mainly because data available in the literature are still scarce in this field. Besides, the absence of specific clinical cut-off values and routine laboratory procedures, particularly for cytokines assessment, makes any conclusion unadvised or reckless. In any case, in the discussion section, we have provided more specific suggestions regarding the combined evaluation of local cytokines and homocysteine and their potential in the management and follow-up of patients with VU. Accordingly, to standardize the procedures and reinforce our observations, we are working on a large multicenter study to identify the potential role of local cytokines, serum homocysteine, and microbiome in VUs.

Reviewer 2 Report

This study compared the level of serum homocysteine and inflammatory cytokines and microbial profiles from the wound fluid between IVU and NVU patients. The result showed levels of several cytokines such as IL-6 IL-17A and TNF-alpha were significantly higher in IVUs than NVUs. Interestingly, bacteria strains isolated from IVU patients show a strong biofilm-forming ability, which might be related to the intense immune response. 

My concerns:

1.     Microbiology tests might also be processed with samples from lesions of NVU patients. It will help to remove the interference from environmental and skin microbiota.

2.     The level of serum homocysteine could be upregulated in several accurate or chronic infections. The complex relationship between serum homocysteine levels and biofilm-producing bacteria should be carefully discussed. 

3.     The quantitative definition of Strong biofilm producers and weak biofilm producers should be described in the Material and methods for convenience. The description should be consistent (Strong biofilm producers or strong 162 biofilm-producers)

Besides anaerobes, fungi should also be considered in the author’s future study. 

Author Response

First, we would like to thank the reviewers for the time spent reviewing our manuscript and the valuable comments and suggestions, which were of great help in improving our work. Accordingly, the manuscript has been revised with new information and additional interpretation of the data. In particular, we have expanded the description of the Biofilm Ring Test and the definition of strong and weak biofilm producers. Besides, we have addressed all specific comments, as seen in the enclosed point-by-point list. The changes from the original submission are highlighted in a track changes version of the document.

Comments and Suggestions for Authors

This study compared the level of serum homocysteine and inflammatory cytokines and microbial profiles from the wound fluid between IVU and NVU patients. The result showed levels of several cytokines such as IL-6, IL-17A, and TNF-alpha were significantly higher in IVUs than NVUs. Interestingly, bacteria strains isolated from IVU patients show a strong biofilm-forming ability, which might be related to the intense immune response. 

My concerns:

Microbiology tests might also be processed with samples from lesions of NVU patients. It will help to remove the interference from environmental and skin microbiota.

Reply: We agree with the reviewer that the possibility of processing samples from lesions of NVU patients may have helped. However, in most cases, cultivation-based microbiology tests performed on NVU were negative. Thus, to have a homogeneous data set, we decided to analyze only samples from IVU. We have also commented, in the discussion section, on the potential issue linked to the culture-based methods. Thus, to overcome these limitations, we are currently working on a large multicenter study to identify the potential role of the microbiota by culture-independent molecular methods in patients with VUs.

The level of serum homocysteine could be upregulated in several accurate or chronic infections. The complex relationship between serum homocysteine levels and biofilm-producing bacteria should be carefully discussed. 

R: We thank the reviewer for pointing this out. Indeed, this is an exciting aspect to analyze. We speculated on a possible role of homocysteine in the methionine cycle and the quorum-sensing (QS) mechanism in regulating biofilm production (lines 333 – 337). Unfortunately, the clinical and scientific evidence is scarce, and they do not allow us to go any further, which is what we really need to do.

The quantitative definition of Strong biofilm producers and weak biofilm producers should be described in the Material and methods for convenience. The description should be consistent (Strong biofilm producers or strong 162 biofilm-producers)

R: We have modified and homologated all the definitions for weak and strong biofilm producers.

Besides anaerobes, fungi should also be considered in the author’s future study. 

R: We thank the reviewer for this comment. Indeed, as mentioned before, we have started an extensive study on patients with VLU to assess the microbiota's contribution (including anaerobes). The possibility of extending our analysis to fungi and potentially also viruses is a positive and constructive suggestion for future applications.